# Nicotine, THC, and Dolutegravir Modulate E-Cigarette-Induced Changes in Addiction- and Inflammation-Associated Genes in Rat Brains and Astrocytes

**DOI:** 10.3390/brainsci13111556

**Published:** 2023-11-07

**Authors:** Jacqueline Renee Kulbe, Lauren Nguyen, Alexandra Anh Le, Anna Elizabeth Laird, Michael A. Taffe, Jacques D. Nguyen, Jerel Adam Fields

**Affiliations:** 1Department of Psychiatry, University of California San Diego, La Jolla, CA 92093, USA; jkulbe@health.ucsd.edu (J.R.K.); lsnguyen@ucsd.edu (L.N.); atl068@ucsd.edu (A.A.L.); alaird@health.ucsd.edu (A.E.L.); mtaffe@health.ucsd.edu (M.A.T.); 2Department of Psychology and Neuroscience, Baylor University, Waco, TX 76706, USA; jacques_nguyen@baylor.edu

**Keywords:** THC, nicotine, antiretroviral drugs, neuroinflammation

## Abstract

E-cigarette use has been marketed as a safer alternative to traditional cigarettes, as a means of smoking cessation, and are used at a higher rate than the general population in people with HIV (PWH). Early growth receptor 2 (EGR2) and Activity-Regulated Cytoskeleton-Associated Protein (ARC) have a role in addiction, synaptic plasticity, inflammation, and neurodegeneration. This study showed that 10 days of exposure to e-cigarette vapor altered gene expression in the brains of 6-month-old, male, Sprague Dawley rats. Specifically, the e-cigarette solvent vapor propylene glycol (PG) downregulated EGR2 and ARC mRNA expression in frontal cortex, an effect which was reversed by nicotine (NIC) and THC, suggesting that PG could have a protective role against NIC and cannabis dependence. However, in vitro, PG upregulated EGR2 and ARC mRNA expression at 18 h in cultured C6 rat astrocytes suggesting that PG may have neuroinflammatory effects. PG-induced upregulation of EGR2 and ARC mRNA was reversed by NIC but not THC. The HIV antiretroviral DTG reversed the effect NIC had on decreasing PG-induced upregulation of EGR2, which is concerning because EGR2 has been implicated in HIV latency reversal, T-cell apoptosis, and neuroinflammation, a process that underlies the development of HIV-associated neurocognitive disorders.

## 1. Introduction

For decades, cigarette smoking has represented a significant health crisis worldwide. Concerted public health efforts in the United States have led to a decline in tobacco use, particularly among teenagers and young adults [1]. However, rates of e-cigarette use have increased since their inception and are popular among adolescents and young adults [2]. E-cigarettes have also been marketed as a safer alternative to traditional cigarettes and as a tool for smoking cessation [3]. E-cigarettes contain a liquid composed of a psychoactive substance (ex: nicotine [NIC] and tetrahydrocannabinol [THC]) and a solvent (ex: propylene glycol) that is vaporized by a heating element and then inhaled by the user. Although the dangers of tobacco and NIC have been well studied, the effect that chronic inhalation of solvent vapors, such as propylene glycol, in combination with psychoactive substances, have on addiction, inflammation, and neurobiology is not well understood.

People with HIV (PWH) use e-cigarettes and other addictive substances at higher rates than the general population and may be more vulnerable to the development of substance use disorders [4,5]. PWH are particularly vulnerable to smoking and psychoactive substance-associated illnesses such as COPD, malignancy, cardiovascular disease, and immune dysfunction even when receiving antiretroviral therapy (ART) [6,7]. PWH on ART are also at risk of developing HIV-associated neurocognitive disorders (HAND), which is in part caused by excessive neuroinflammation [8]. Understanding the effects of e-cigarettes on addiction and neuroinflammatory pathways is important to the general population of e-cigarette users and is of special relevance to people with PWH whose brains have higher baseline levels of neuroinflammation and may be more vulnerable to insult.

Recent studies suggest that e-cigarettes may affect the brain. E-cigarette aerosols with NIC affect neuron and astrocyte function and neurotransmitter processing in mice [9,10]. Another study showed that e-cigarettes impair short- and long-term memory and cause reductions in brain-derived neurotrophic factors in rats [11]. NIC alone binds the nicotinic acetylcholine receptor and acute NIC use has been associated with improved hippocampus-dependent learning, memory, and attention. Contrary to this, chronic NIC use was associated with depressed hippocampus-dependent learning [12]. However, few studies have delineated how NIC, THC, or ART modify e-cigarette-induced changes in gene expression in the frontal cortex.

In recent years, RNA sequencing (RNAseq) and subsequent analyses of the transcriptome have empowered researchers to gain insight into how global gene expression is altered in response to a given stimulus or set of stimuli. Pairing transcriptomics unbiased search for changes in gene expression with traditional approaches such as real-time polymerase chain reaction (RT^2^PCR) and Western blotting is proving to be an efficient way to discover novel gene expression networks and hypotheses.

The goal of this study was to utilize RNAseq technology and transcriptome analyses to investigate the effects of propylene glycol (PG), nicotine (NIC), and THC on gene expression in the brain in an unbiased way. We next selected two important genes (EGR2 and ARC) deemed from the literature to be relevant to PWH and addiction from the transcriptomics analyses for downstream validation mRNA using RT^2^PCR and Western blot, respectively. Lastly, we used qRT^2^PCR to investigate the expression of these genes in cultured astrocytes exposed to PG, NIC, and THC. The effects of anti-retroviral dolutegravir (DTG) in combination with PG, NIC, and THC on astrocytes were also assessed due to its relevance to PWH.

## 2. Methods

### 2.1. Animal Model

Male Sprague Dawley rats (Harlan/Envigo, Livermore, CA, USA) were housed in humidity- and temperature-controlled (23  ±  2 °C) environments on reversed 12:12 h light: dark cycles. All procedures were conducted under protocols approved by the Institutional Care and Use of the University of California, San Diego (Institutional Animal Care and Use Committee protocol: S19029).

### 2.2. Rat Exposure to E-Cigarette Vapor Delivery Apparatus

First, 6-month old, male, Sprague Dawley rats (*n* = 6/group) were placed into vaporization chambers for 30 min sessions twice per day for 10 days and exposed to room air, propylene glycol (PG, 100%), PG + nicotine (NIC 30 mg/mL), or PG + THC (200 mg/mL). Two 30 min sessions were chosen in an attempt to model the multiple per day e-cigarette use seen per day in humans. Nicotine dependence has been shown to develop in as little as 7 to 14 days [13,14]. Vapor was delivered through a vapor inhalation system designed to deliver psychoactive substance vapor as previously described [15,16,17,18,19]. Nicotine and propylene glycol were obtained from Sigma-Aldrich (St. Louis, MO, USA) and THC was provided by the U.S. National Institute on Drug Abuse.

### 2.3. RNA Isolation, Library Prep and Sequencing

RNA (*n* = 6/group) was extracted from frontal cortex Qiagen RNeasy Lipid Tissue Kit per manufacturer’s instructions (Qiagen, Germantown, MD, USA; cat no. 74804). The mRNA library was generated using Illumina^®^ Stranded mRNA Prep kit (San Diego, CA, USA, cat no. 20040532) and sequenced on a NovaSeq 6000 Sequencing System. Analysis of differential gene expression was performed by processing FASTQ files with DRAGEN RNA app and aligning to the Rattus norvegicus genome using the DRAGEN Differential Expression app on Illumina’s BaseSpace cloud software (v 4.2.4).

### 2.4. In-Vitro Studies of Astrocytes

The C6 rat glioma cell line (ATCC cat# CCL-107) was used here for their astroglia characteristics. Astroglia are implicated in frontal cortex function and relevant to addiction and HIV-associated neurocognitive disorders. C6 cultures were grown in DMEM with 5% FBS at 37 °F and 5% CO_2_. C6 rat astrocytes were split into 12 well plates at 5 × 105 cells/well. Cells were treated with various combinations of vehicle (DMEM with 5% FBS), propylene glycol (50 mM, 100 mM, or 150 mM, *n* = 3/group/dose response), Nicotine (10 μM), THC (10 μM), or Dolutegravir (DTG) (200 nM) for 6 h or 18 h and then processed for RNA (*n* = 9–15/group).

### 2.5. Real-Time Reverse Transcription Polymerase Chain Reaction

6 h or 18 h following treatments C6 rat astrocytes were washed with PBS and RNA was extracted and reverse transcribed as previously described [20]. Gene expression assays were performed and analyzed as previously described [21] using primers specific to EGR2 (Taqman, ThermoFisher Scientific, Carlsbad, CA, USA; cat no. Rn00571208_g1), ARC (Taqman, Rn00571208_g1), and ActB (Taqman, Rn00667869_m1).

### 2.6. Immunoblot of Human Brain Specimens

Frontal cortex tissues (*n* = 6/group) from rat brains were homogenized and Western blot and analysis per performed as previously described [20,21,22]. Antibodies were used as follows: ARC (ARC/ARG3 polyclonal; Proteintech, Chicago, IL, USA; cat no. 16290-1-AP; 1:500), EGR2 (EGR2 polyclonal; Proteintech; cat no. 13491-1-AP; 1:500), β-actin (ACTB; Sigma-Aldrich, cat. no. A5441; 1:2000), species-specific IgG conjugated to HRP (American Qualex, San Clemente, CA, USA; cat. no. A102P5; 1:5000).

### 2.7. Statistical Analysis

Statistical analysis was determined by one-way ANOVA with Tukey’s post hoc test using GraphPad Prism (v 10.0.2) with a *p* < 0.5 considered significant.

## 3. Results

### 3.1. Differential Gene Expression (Figure 1)

Following exposure to Air, PG, PG + NIC, or PG + THC (*n* = 6/group), RNA was extracted from whole brain lysate, sequenced, and differential gene expression was analyzed as above. Fourteen genes were significantly (*p* < 0.05) differentially expressed between PG and Air groups, including Egr1, ARC, Rsrp1, Junb, Emcn, Miiip, Trib1, Hspa1b, Egr2, Abca7, Slc39a&, Egr4, Fos, and Dusp6. Between the PG and NIC groups, 13 genes were significantly (*p* < 0.05) differentially expressed including Gemin6, Hspa1b, Zfp180, Rsrp1, Junb, Per1, Egr4, Nr4a3, Ift122, Errfil1, Egr2, Cttn, and AABR07065531.5. Between NIC and THC three genes were significantly (*p* < 0.05) differentially expressed, including Il1r1, Tgm2, and Ebna1bp2.

Analysis identified several Early Growth Response Factor genes that were differentially expressed between PG versus Air and PG versus NIC, including EGR2 (Early Growth Receptor 2 gene) which was downregulated in PG compared to both Air and NIC. The intermediate-early gene ARC (Activity Regulated Cytoskeleton-Associated gene) showed high levels of expression in both PG and AIR and was also downregulated in PG. 

**Figure 1 brainsci-13-01556-f001:**
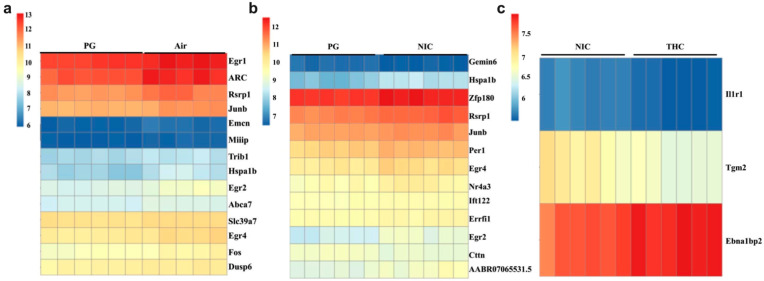
E-cigarette vapor, nicotine, and THC alter gene expression in rat frontal cortex. Heatmaps demonstrating significant differential gene expression (*p* < 0.05) for mRNA extracted and sequenced from whole brain lysate of rats exposed to e-cigarette vapor for 10 days. (**a**) PG (100% propylene glycol) vs. AIR (air), (**b**) PG vs. NIC (PG + Nicotine 30 mg/mL), (**c**) NIC vs. THC (PG + THC 200 mg/mL).

### 3.2. Frontal Cortex mRNA and Protein Expression of EGR2 and ARC (Figure 2)

Quantitative RT-PCR was used to determine the fold changes for EGR2 and ARC mRNA in frontal cortex of rats exposed to air (control), PG (vehicle), PG + THC, and PG + NIC (*n* = 5–6/group). For EGR2, one-way ANOVA revealed a significant decrease in PG compared to AIR (*p* = 0.01), THC (*p* = 0.0359), and NIC (*p* = 0.0007) (Figure 2a). However, EGR2 mRNA levels do not significantly differ between AIR, THC, and NIC, suggesting that THC and NIC negate the effect PG has on the downregulation of EGR2. These data are consistent with the differential expression results obtained from whole brain lysates which showed that EGR2 was downregulated in PG compared to AIR and NIC (Figure 1a,b) and consistent with Figure 1c which shows that EGR2 was not identified as being differentially expressed between NIC and THC. Western blot did not identify significant differences in EGR2 protein expression in the frontal cortex of rats exposed to AIR, PG, THC, or NIC (Figure 2d). However, the PG group had the lowest mean EGR2 protein expression (approximately 10% decrease compared to AIR). These data suggest that while a 10-day exposure to these substances is sufficient to induce mRNA, it may not be a long enough period of time to alter protein expression significantly.

ARC showed a similar pattern in frontal cortex as EGR2 with PG having the lowest mean fold change (approximately 10% decrease compared to AIR) and THC and NIC groups being more similar to air suggesting PG downregulates ARC mRNA and this downregulation is reversed by THC and NIC. However, these results were not statistically significant (Figure 2b). Similarly, significant changes were not detected in ARC protein expression; however, the PG group had the lowest mean level of protein expression amongst all groups (Figure 2e).

**Figure 2 brainsci-13-01556-f002:**
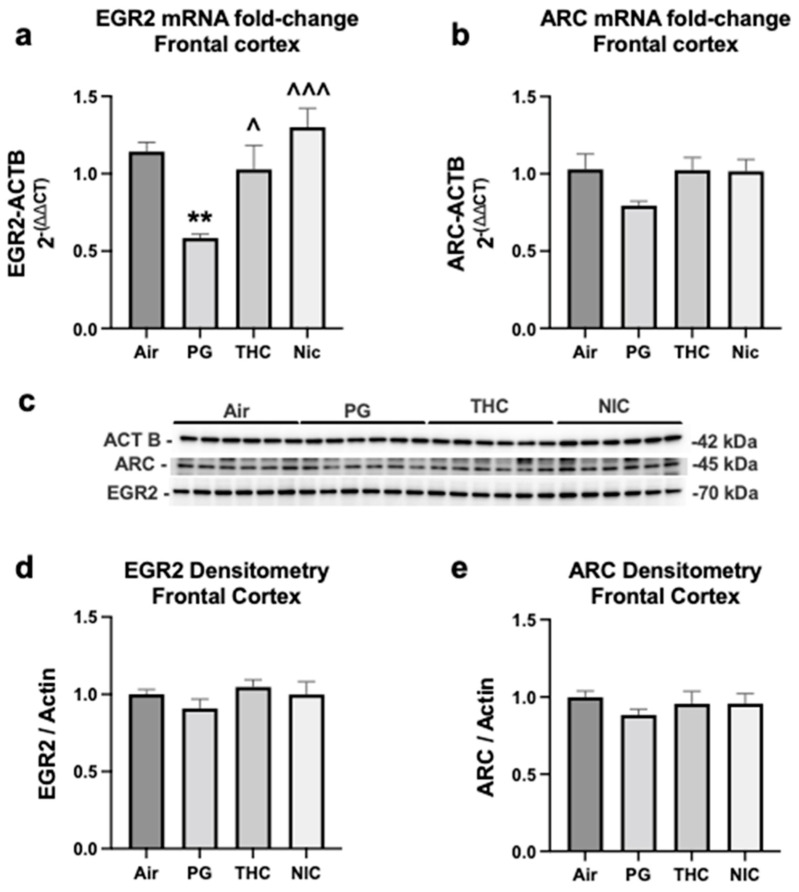
Propylene glycol (PG) decreases EGR2 and ARC mRNA and protein expression in frontal cortex of rats exposed to e-cigarette vapor for 10 days and these changes are reversed by THC and nicotine. (**a**) Fold change in EGR2 and (**b**) ARC mRNA normalized to ACTB. (**c**) Immunoblot of frontal cortex with an antibody specific for EGR2, ARC, and Actin. Quantification of immunoblot for (**d**) EGR2 and (**e**) ARC band intensity normalized to actin. AIR = Air, PG = 100% propylene glycol, THC = PG + THC 200 mg/mL. NIC = PG + nicotine 30 mg/mL. Mean ± SEM. Statistical significance was determined by one-way ANOVA, post hoc Tukey’s. ** *p* < 0.01 vs. Air; ^ *p* < 0.05, ^^^ *p* < 0.001 vs. PG.

### 3.3. Time Course and Dose–Response of PG-Induced mRNA Expression in Rat Astrocytes

A dose–response (*n* = 3/group) of PG (5 mM, 50 mM, 150 mM) was conducted in vitro to evaluate changes in mRNA expression for EGR2 and ARC in cultured C6 rat astrocytes. At 18 h, 150 mM of PG significantly increased EGR2 mRNA (*p* = 0.0187) expression by 20% compared to control (Figure 3a) and significantly increased ARC mRNA expression compared to control (*p* = 0.0024) and 5 mM (*p* = 0.0080) and 50 mM doses (*p* = 0.0055) (Figure 3b). 150 mM PG did not result in increases in cytotoxicity. Therefore, 150 mM PG was used for further in vitro experiments. Next, ERG2 and ARC mRNA expression were evaluated in C6 rat astrocytes that had been treated with 150 mM PG for either 6 h or 18 h (*n* = 6–8/group). Similar to in vivo data, at 6 h EGR2 mRNA was significantly decreased compared to vehicle (Figure 3c) but then showed a significant increase at 18 h when compared to vehicle or 6 h. For ARC, mRNA was significantly increased at both 6 h and 18 h compared to vehicle with the 6 h timepoint also being significantly elevated compared to 18 h (Figure 3d).

### 3.4. PG, NIC, THC, and DTG-Induced EGR2 and ARC mRNA Expression in Rat Astrocytes

C6 Rat astrocytes were treated with PG (150 m), NIC (10 μM), THC (10 μM), or dolutegravir (200 ng/mL) for 18 h in the following combinations (*n* = 9–15/group): Vehicle (media), PG, PG + NIC, PG + THC, DTG, PG + DTG, PG + THC + DTG, PG + NIC + DTG. For EGR2 (Figure 4a), PG significantly increased (*p* = 0.0005) mRNA expression by 55% compared to control at 18 h, an effect that was reversed by the addition of nicotine (*p* = 0.2546; vehicle vs. PG + NIC). The addition of THC further elevated EGR2 mRNA by 20% compared to PG alone and was significantly increased compared to control (*p* = 0.016). DTG alone had minimal effect on EGR2 mRNA expression compared to vehicle (*p* = 0.9927). PG + DTG was significantly elevated compared to DTG (*p* = 0.0107) indicating that PG-induced increases in EGR2 mRNA expression occur in the presence of DTG. However, EGR2 mRNA expression was 13% but non-significantly increased (*p* = 0.8465) in PG + NIC + DTG compared to PG + NIC group, suggesting the addition of DTG may interfere with the ability of NIC to attenuate PG-induced increases in EGR2 mRNA expression. Although non-significant, a similar pattern was seen for ARC mRNA.

## 4. Discussion

Our results indicate that ten days (2 × 30 min per day) of exposure to PG ± NIC or THC is sufficient to alter gene expression in the brains of 6-month-old, male, Sprague Dawley rats. RNA sequencing revealed several differentially expressed genes between treatment groups including several early growth response genes (EGR1, EGR2, EGR4), with EGR2 being downregulated in PG-exposed animals compared to AIR or PG + Nicotine (Figure 1a,b) and ARC being downregulated in PG compared to AIR (Figure 1a). Similarly, EGR2 mRNA was significantly decreased, and ARC mRNA was non-significantly decreased in PG-exposed frontal cortex (Figure 2a,b). These effects were reversed by the addition of NIC or THC (Figure 2a,b). Typically, it is unwise to draw conclusions from non-significant data. However, because the ARC mRNA levels show a similar overall trend to that of ERG2 we believe that these trends deserve discussion. Therefore, the hypothesis and implication regarding both EGR2 and ARC changes are discussed below.

PG is a food additive generally considered safe by the FDA. PG vapor has been found to contain dangerous chemicals such as formaldehyde and acrolein but animal studies evaluating the effects of PG vapor generally indicate it to be non-toxic [23]. However, the majority of these studies focus on the respiratory tract and other peripheral organ systems. Our results indicated that PG vapor is sufficient to induce changes in gene expression within the brains of rats after only 10 days of exposure. Alterations in gene expression for our validated genes, EGR2 and ARC, did not translate to changes in protein expression (Figure 2c,d) as assessed by Western blot. Although it is possible that the 10-day exposure was not long enough for alternations in mRNA expression to translate to alterations in protein expression, given the previously rapid induction of ARC and EGR2 seen in models of psychoactive substance administration [24,25], it is also possible that alterations in post-translational modification are occurring. Post-translational modifications are known to occur and regulate both EGR2 and ARC function [26]. Future studies focusing on evaluation of post-translational modifications as well as upstream modulators are needed.

EGR2 and ARC were chosen for further evaluation because they both have a role in addiction, inflammation, and neurodegeneration. EGR2 is a transcription factor characterized as an immediate-early gene and its transcription is rapidly induced by a variety of extracellular stimuli [27]. ARC is also an immediate-early gene and encodes for activity-regulated cytoskeletal-associated protein. Both are important in synaptic plasticity [28,29,30,31] which is an underlying mechanism for both addiction [32] and the memory and cognitive impairment seen in neurodegenerative disorders such as Alzheimer’s Disease and HAND [33].

Alterations in EGR2 and ARC are found in addiction models [25,31,34,35,36]. Changes in ARC are induced rapidly, in as little as 30 min [24], following psychoactive substance administration, with increases sustained chronically [31,37], and decreases induced by psychoactive substance withdrawal [31,37]. EGR-2 is also induced within hours of psychoactive substance administration, including nicotine administration [25], with increases sustained chronically depending on brain region and treatment paradigm [38,39]. EGR2 and ARC are induced or upregulated by psychoactive substances including cocaine, methamphetamine, alcohol, heroin, and nicotine [25,31,35]. CBD also induces EGR2 expression [40]. Therefore, it is unsurprising that in the brains of rats exposed to PG, nicotine was able to reverse PG-induced decreases in EGR2 and ARC mRNA expression (Figure 2a,b). Nor is it surprising that THC also reversed PG-induced decreases in EGR2 expression. However, the ability of THC to reverse PG-induced decreases in ARC mRNA expression is in contrast to a previous study in which THC decreased ARC protein expression [41]. However, that study was conducted in adolescent rats, THC was administered systemically without PG exposure, and significant differences were only found in the prefrontal cortex of female rats [41].

It is unclear how inhalation of PG vapor causes decreases in EGR2 and ARC mRNA expression and why the effect is more robust in EGR2. It is possible that the modulation of EGR2 and ARC are not induced directly by PG, NIC, or THC themselves, but that these substances are acting through upstream transcription factors or kinases. EGR2 itself is a transcription factor capable of binding ARC [42], which may explain the more robust changes seen in EGR2 compared to ARC. ARC is regulated by a variety of upstream factors including cAMP response element-binding protein (CREB), Myocyte Enhancer Factor 2 (MEF2), and Serum response factor (SRF) [43], as well as the kinase Extracellular Signal-regulated Kinases (ERK) [44]. Both ARC and EGR2 have also been associated with Nrf2 (Nuclear factor erythroid-2 related factor 2), a transcription factor involved in the regulation of antioxidant and inflammatory responses. Specifically, knockdown of ARC has been demonstrated to inhibit Nrf2 expression [45]. And is increased in parallel to Nrf2 in transgenic animals less prone to oxidative stress [46], whereas parallel increases in EGR2 and Nrf2 are associated with improved retention memory [47]. Interestingly, CBD has been shown to modulate Nrf2 and its upstream regulator (Nuclear factor kappa-light-chain-enhancer of activated B cells) NFkB [48], which also regulates EGR2. Further studies are needed to evaluate the role of upstream transcription factors in the modulation of EGR2 and ARC by PG, NIC, or THC. It is also unclear what the neurobiological implications of these findings might be. It is possible that PG-induced decreases in EGR2 could have a protective effect against nicotine- or THC-induced increases in EGR2 expression, lowering the overall risk of nicotine or cannabis dependence. However, that conclusion cannot be made based on the current work because behavioral studies were not conducted. However, in support of this hypothesis, a previous study demonstrated that the addition of PG decreased the intracranial self-administration threshold of high-dose nicotine in rats [49]. On the other hand, as discussed further below, because decreases in ARC correlate with both neuroinflammation and cognitive impairment [33], it is possible that PG-induced decreases in ARC could enhance neuroinflammation and cognitive impairment in e-cigarette users who are more vulnerable to those pathologies such as those with HIV.

The importance of EGR2 and ARC is not limited to addiction. Immunologically, EGR2 is involved in B- and T-cell development [50], macrophage polarization [51], and T-cell suppression [52]. ARC is involved in peripheral immune cell migration, T-cell activation [53], and upregulated following prenatal inflammation [54]. In specific regard to HIV, upregulation of early growth response proteins, including EGR2, can lead to HIV reactivation and latency reversal [55] and binding of the HIV tat protein to EGR2 can enhance T-cell apoptosis [56]. Neurologically, in addition to their roles in synaptic plasticity, EGR2 is involved in hindbrain development, peripheral nerve myelination, heritable peripheral neuropathies, and Huntington’s Disease [57,58,59]. Alterations in ARC expression and post-translational modifications of ARC occur both in genetic neurodevelopmental disorders and Alzheimer’s Disease [30]. In fact, ARC enhances the cleavage of amyloid precursor protein (APP) to Aβ peptide [30] which is known to accumulate in both Alzheimer’s and HAND. Less is known about the role EGR2 and ARC have in neuroinflammation. Although EGR2 is upregulated in Aβ plaque-associated microglia [60], neuroinflammatory stimuli lead to upregulation of EGR2 in neurons [61], microglia, and astrocytes [62], and neuroinflammation alters activity-dependent ARC synthesis in hippocampal neurons [33].

Similar to the in vivo findings (Figure 2), PG downregulated EGR2 mRNA expression in cultured C6 rat astrocytes at 6 h but upregulated EGR2 at 18 h (Figure 3c). ARC was significantly upregulated at both 6 h and 18 h (Figure 3d). The majority of ARC studies focus on neuronal expression of ARC, making the finding that PG is able to upregulate ARC mRNA in C6 rat astrocytes quite novel.

It is unclear why ERG2 was initially downregulated in C6 rat astrocytes, however, the discrepancy between the in vivo PG-induced downregulation of EGR2 and ARC and the in vitro upregulation of EGR2 and ARC is possibly due to a cell-specific effect and the in vivo environment contains many cell-types interacting together to maintain homeostasis. Alternatively, although the concentration of PG was chosen to ensure a robust in vitro response in order to study the effects of the addition of NIC, THC, and DTG, we concede that a 50–150 mM concentration of PG is unlikely comparable to physiologic levels following e-cigarette exposure which are likely not to be more than in the micromolar range [63]. Therefore, modulation of EGR2 and ARC in cultured rat astrocytes at 6 h and 18 h may be the result of a stress response to a non-physiologic chemical or osmotic insult rather than a PG-specific effect. Reassuringly, these concentrations of PG did not induce cytotoxicity. Both EGR2 and ARC are known to be upregulated under stressful conditions or alterations in metabolic activity [64]. Although both EGR2 and ARC are induced rapidly following stimulus [24,25,62], we suspect that the different pattern seen between EGR2 and ARC at 6 h and 18 h is likely due to temporal differences in expression induction between EGR2 and ARC, with decreases in ARC at 18 h possibly being the result of other homeostatic mechanisms. Future studies are needed to evaluate more physiologically relevant concentrations of PG, and earlier and later time points. However, we find the ability of THC to further enhance this effect and NIC to reduce EGR2 to levels similar to control particularly intriguing. Additionally, future in vivo studies evaluating the effects of e-cigarettes on individual cell types, including neuroimmune cells such as astrocytes and microglia are needed.

Similar to the in vivo studies (Figure 2), NIC reversed PG-induced alterations in EGR2 gene expression. In this case, reversing PG-induced upregulation of EGR2 (Figure 4a). ARC expression changes were subtle and non-significant but followed a similar overall pattern to that of EGR2 (Figure 4). PG vapor has been previously shown to increase pro-inflammatory cytokine expression [65]. It is possible that in this model, PG is acting as a pro-inflammatory stimulus, leading to upregulation of EGR2, and that nicotine is having an anti-inflammatory effect [66]. To test this hypothesis, future studies evaluating cytokine production and astrocyte morphology in vivo and in vitro will be needed.

The ability of PG to upregulate EGR2 in cultured astrocytes, possibly due to an underlying neuroinflammatory mechanism, is particularly concerning to THC e-cigarette users because THC did not reverse PG-induced upregulation of EGR2 mRNA in cultured C6 rat astrocytes, but instead furthered increased EGR2 mRNA levels compared to PG alone (Figure 4a). However, it should be noted that variability was high in this group. It is unclear why nicotine but not THC was able to reduce PG-induced upregulation of EGR2 mRNA given the current interest in the anti-inflammatory properties of cannabis [67]. However, the anti-inflammatory properties of cannabis are most likely due to non-THC cannabinoids [68]. In regard to PWH, the ability of PG to upregulate EGR2 in cultured astrocytes is alarming due to the possibility that EGR2 is involved in HIV latency reversal and CD4 apoptosis [55,56] and that chronic neuroinflammation contributes to the development of HAND [8]. However, the differences in the effect of PG on EGR2 expression in rat brains (decreased expression) versus in C6 rat astrocytes (decreased at six hours and increased at 18 h) may indicate a differential of PG on astrocytes in the in vivo environment. Bystander brain cell types (neurons, microglia, oligodendrocytes) may harbor the observed changes in EGR2 expression or may modulate the effects of PG on astrocytes in vivo. Moreover, the differences in the in vivo versus in vitro studies may be driven by acute versus chronic exposure to PG, NIC, and THC. Further, Dolutegravir, an HIV integrase inhibitor found in the common HIV antiretroviral medication Tivicay, reversed the effect NIC had on decreasing PG-induced upregulation of EGR2 (Figure 4a), suggesting that for PWH, DTG has the ability to further modulate e-cigarette-induced alterations in gene expression in ways that we do not yet understand. Future studies focusing on the interaction between e-cigarettes and antiretrovirals in models of HIV are needed.

Limitations of the study: This study did not investigate the combinatorial effect of DTG with Nic or THC in the absence of PG, which is relevant to PWH smoking cannabis in traditional combustion of marijuana cigarettes or pipes. While this is important for understanding the effects of NIC and THC on the brain in PWH and PWoH, the studies here focused on e-cigs which generally use a vehicle for NIC or THC, in this case PG. This study does not fully address the differences in gene expression in response to PG, NIC, and THC in rat brains versus C6 rat astrocytes. Additionally, although nicotine dependence in rats had been shown to develop in as little as 7 days during nicotine vaporization, and one rat day is equivalent to approximately 30 human days, now that this study has demonstrated acute gene changes following 10 days of e-cigarette exposure, more in-depth studies are necessary to determine the effects of acute versus chronic exposure to e-cigarettes and also to understand how PG affects EGR2, ARC and other gene expression in the brain.

## 5. Conclusions

E-cigarette use has been marketed as a safer alternative to traditional cigarettes, as a means of smoking cessation, and is used at a higher rate than the general population in PWH. EGR2 and ARC have a role in addiction, synaptic plasticity, inflammation, and neurodegeneration. Overall, this study showed that 10 days of exposure to e-cigarette vapor was sufficient to alter gene expression in the brains of 6-month-old, male, Sprague Dawley rats. Specifically, PG significantly downregulated EGR2 and non-significantly downregulated ARC mRNA expression in frontal cortex, an effect which was reversed by NIC and THC. However, this did not translate to alterations in protein expression. PG-induced decreases in EGR2 mRNA in rat frontal cortex may suggest that PG could have a protective role against NIC and cannabis dependence. However, in vitro, PG upregulated EGR2 and ARC mRNA expression at 18 h in cultured C6 rat astrocytes, suggesting that PG may have neuroinflammatory effects. The finding that PG can induce ARC expression in astrocytes is novel because previous ARC studies have focused on neuronal expression. PG-induced significant upregulation of EGR2 and non-significant upregulation of ARC mRNA in cultured C6 rat astrocytes was reversed by NIC but not THC. In fact, THC enhanced the PG-induced increases in EGR2, which may have concerning implications for THC e-cigarette users. The HIV antiretroviral DTG reversed the effect NIC had on decreasing PG-induced upregulation of EGR2. This is concerning because EGR2 has been implicated in HIV latency reversal, T-cell apoptosis, and neuroinflammation, a process that underlies the development of HAND. Future in vitro and in vivo studies investigating the effects of e-cigarettes on addiction, neuroinflammation, and neurodegeneration are needed and should include HIV models, evaluation of the interaction between e-cigarettes and antiretrovirals, and focus on neuroimmune cells such as astrocytes and microglia.

## Figures and Tables

**Figure 3 brainsci-13-01556-f003:**
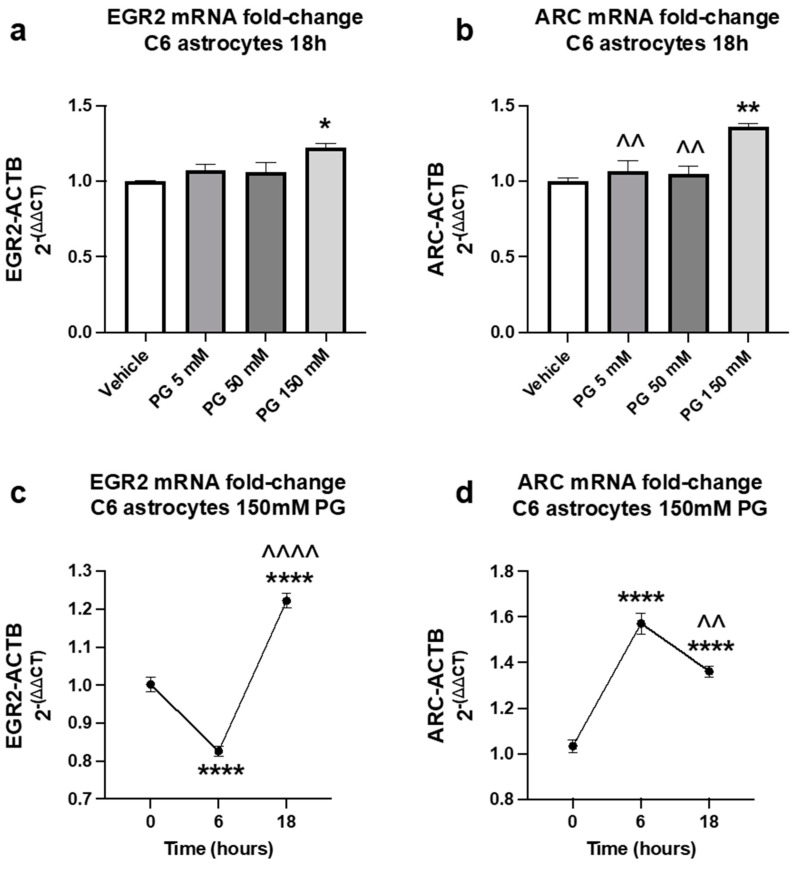
PG (propylene glycol) alters EGR2 and ARC mRNA expression at 6 h and increases EGR2 and ARC mRNA expression in a dose-dependent manner at 18 h in cultured C6 rat astrocytes. Fold change in (**a**) EGR2 and (**b**) ARC mRNA levels normalized to ACTB following 18 h treatment with 5 mM, 50 mM, or 150 mM PG. Fold in of (**c**) EGR2 and (**d**) ARC mRNA levels normalized to ACTB following 6 h or 18 h treatment with 150 mM PG. Mean ± SEM. Statistical significance was determined by one-way ANOVA, post hoc Tukey’s. (**a**,**b**) * *p* < 0.05, ** *p* < 0.01 vs. vehicle; ^^ *p* <0.01 vs. 150 mM PG (**c**,**d**) **** *p* < 0.0001 vs. 0 h; ^^ *p* < 0.005 vs. 6 h, ^^^^ *p* < 0.00001 vs. 6 h.

**Figure 4 brainsci-13-01556-f004:**
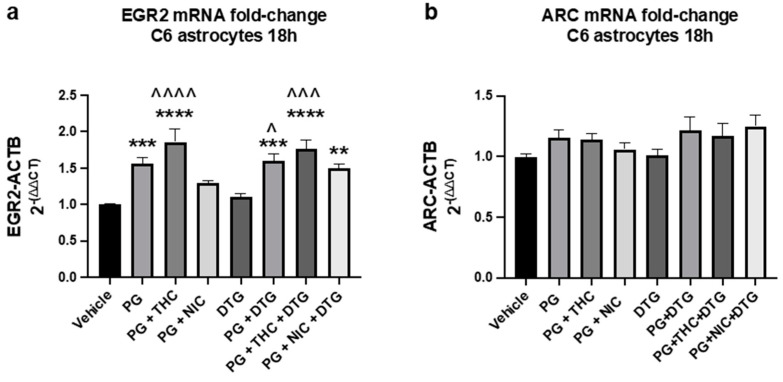
PG (propylene glycol) increases EGR2 and ARC mRNA expression in C6 rat astrocytes at 18 h and is reversed by Nicotine (NIC), an effect that is lost with the addition of dolutegravir (DTG). Fold change in (**a**) EGR2 and (**b**) ARC mRNA levels normalized to ATCB following 18 h treatment with Vehicle, PG (150 mM), PG + THC (10µM), PG + NIC (10µM), DTG (200 nM), DTG, PG + DTG, PG + THC + DTG, or PG + NIC + DTG. Mean ± SEM. Statistical significance was determined by one-way ANOVA, post hoc Tukey’s. ** *p* < 0.01, *** *p* <0.001, **** *p* < 0.0001 vs. vehicle, ^ *p* <0.05, ^^^ *p* < 0.001, ^^^^ *p* < 0.0001 vs. DTG.

## Data Availability

All data will be available upon reasonable request.

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
