# Peer review of "Nicotine, THC, and Dolutegravir Modulate E-Cigarette-Induced Changes in Addiction- and Inflammation-Associated Genes in Rat Brains and Astrocytes"

_brainsci, 2023, doi:10.3390/brainsci13111556_

Round 1
Reviewer 1 Report
Comments and Suggestions for Authors
The work evaluated the effect of PG,NIC,THC on the expression of Early growth receptor 2 (EGR2) and Activity Regulated Cytoskeleton Associated Protein (ARC), proteins associated with addiction, synaptic plasticity, inflammation, and neurodegeneration in animals. In C6 rat astrocytes they evaluated the effect of PG,NIC and THC, but with an antiviral drug DGT. The results in animals, PG downregulated EGR2 and ARC mRNA expression in frontal cortex, an effect which was reversed by nicotine (NIC) and THC, suggesting a protective effect. While cultured C6 rat astrocytes in-vitro, PG upregulated EGR2 and ARC mRNA expression suggesting a neuroinflammatory effect of PG. An interesting manuscript, however I have some questions
Why didn't they administer DGT combined with NIC, THC and PG alone in cell culture?
It would be interesting to do it in animals and contrast the effect in both models.
Insert into the discussion the contrast of the effect of PG in both models
Minor comments
Page 3 line 98, the Word of is repeat…” various combinations of of vehicles “
Insert the "n" into the method
In the method, abbreviate dolutegravir, according to the use in the figures (DTG)
Improve the resolution of figure 1
In the introduction they do not mention which molecules they are going to evaluate, the information is in the abstract, introduce this information in the introduction
In the introduction they only justify the genes EGR2 and ARC, but and the other genes as shown in the results of figure 1
Author Response
R1, comments:
- Why didn't they administer DGT combined with NIC, THC and PG alone in cell culture? It would be interesting to do it in animals and contrast the effect in both models.
Response: This is an interesting and insightful suggestion by R1. We have addressed this question in the Discussion section (page 10; lines #362-368 & lines #374-385) with the following text:
“However, the differences in the effect of PG on EGR2 expression in rat brains (decrease expression) versus in C6 rat astrocytes (decreased at six hours and increased at 18 hours) may indicate a differential of PG on astrocytes in the in vivo environment. Bystander brain cell types (neurons, miroglia, oligodendrocytes) may harbor the observed changes in EGR2 expression or they modulate the effects of PG on astrocytes in vivo. Moreover, the differences in the in vivo versus in vitro studies may be driven by acute versus chronic exposure to PG, NIC, and THC.”
“Limitations of the study: This study did not investigate the combinatorial effect of DTG with NIC or THC in the absence of PG, which is relevant to PWH smoking cannabis in traditional combustion of marijuana cigarettes or pipes. While this is important for understanding the effects of NIC and THC on the brain in PWH and PWoH, the studies here focused on e-cigs which generally use a vehicle for NIC or THC, in this case PG.”
R1 Minor comments:
Page 3 line 98, the Word of is repeat…” various combinations of of vehicles “
Response: corrected
Insert the "n" into the method
Response: The n’s per group have been inserted into the methods section
In the method, abbreviate dolutegravir, according to the use in the figures (DTG)
Response: corrected
Improve the resolution of figure 1
Response: A higher resolution image for figure 1 has been added (page 4)
In the introduction they do not mention which molecules they are going to evaluate, the information is in the abstract, introduce this information in the introduction
In the introduction they only justify the genes EGR2 and ARC, but and the other genes as shown in the results of figure 1
Response: This is an important point, which we have now addressed in the Introduction section (page 2, line numbers #59-73)
Response: “In recent years, RNA sequencing (RNAseq) and subsequent analyses of the transcriptome have empowered researchers to gain insight into how global gene expression is altered in response to a given stimulus or set of stimuli. Pairing transcriptomics unbiased search for changes in gene expression with traditional approaches such as real-time polymerase chain reaction (RT2PCR) and western blotting is proving to be an efficient way to discover novel gene expression networks and hypotheses.
The goal of this study was to utilize RNAseq technology and transcriptome analyses to investigate the effects of propylene glycol (PG), nicotine (NIC), and THC on gene expression in the brain in an unbiased way. We next selected two important genes (EGR2 and ARC) deemed from the literature to be relevant to PWH and addiction from the transcriptomics analyses for downstream validation mRNA using RT2PCR and western blot, respectively. Lastly, we used qRT2PCR to investigate the expression of these genes in cultured astrocytes exposed to PG, NIC, and THC. The effects of anti-retroviral dolutegravir (DTG) in combination with PG, NIC, and THC on astrocytes was also assessed due its relevance to PWH. “
Reviewer 2 Report
Comments and Suggestions for Authors
The paper presented to me for review deals with a very interesting issue, which is E-cigarettes use has been marketed as a safer alternative to traditional cigarettes. In the paper, the authors focused on the potential pathogenetic mechanisms of the impact of E-cigarettes used by HIV-infected individuals. The work is well planned, the rationale for the study is appropriate. The methodology of the work on the animal model is not questionable and is well described. I have no major comments only a few minor ones, which, supplemented, can increase the merit of the work:
1. it would be worthwhile to address in the introduction the potential mechanisms of nicotine/tobacco effects on cognitive function based on: PMID: 34208753
2. given the fact that this is an experimental study, it would be worthwhile to complete the exact number of approval of the local bioethics committee
Author Response
- it would be worthwhile to address in the introduction the potential mechanisms of nicotine/tobacco effects on cognitive function based on: PMID: 34208753
Response: This is an important point, which we have now addressed in the Introduction section (page 2, line numbers #53-56)
“NIC alone binds the nicotinic acetylcholine receptor and acute NIC use has been associated with improved hippocampus-dependent learning, memory and attention. Contrary to this, chronic NIC use was associated with depressed hippocampus-dependent learning”
- given the fact that this is an experimental study, it would be worthwhile to complete the exact number of approval of the local bioethics committee
Response: The IACUC protocol number S19029 has been added (pg 2, line 79)
Reviewer 3 Report
Comments and Suggestions for Authors
The manuscript represents an attempt to examine effects of exposure of young adult rat forebrain and cultured astrocytes to components of electronic cigarette aerosol and to draw conclusions about the toxicology.
1. Abstract lines 17-24, line 358: Certainly astrocytes are not the only type of cell in the brain. However, the finding of downregulation of EGR2 and ARC mRNA in forebrains of whole animals exposed to propylene glycol versus upregulation of EGR2 and ARC mRNA in cultured astrocytes raises yellow flags. This should be better addressed at least in the discussion. I note the differences in concentrations of the different substances to which cultured astrocytes were exposed in vitro in the medium. Dolutegravir exposure is 200 nM, Nicotine and THC 10 µM (0.01 mM), but propylene glycol 50 to 150 mM (0.05 to 0.15 M). The propylene glycol concentrations reach the concentration of salt in blood. It is unlikely that propylene glycol would reach these concentrations in the extracellular milieu surrounding brain cells of any kind. Propylene glycol reaches the blood but only after a first pass metabolism in the lungs where an NAD+ dependent alcohol dehydrogenase converts much propylene glycol to lactaldehyde. The authors must comment at least in the discussion that the propylene glycol exposures to cultured astrocytes were very high concentrations and as a consequence, the astrocyte results could be artifactual or a consequence of a cellular chemical or osmotic stress response rather than a specific physiological response to propylene glycol exposure.
2. Lines 202-214 and Figure 3C: The initial changes in EGR2 and ARC mRNA may well represent a response such as described in my previous comment from which the cells are undergoing recovery at 18 hours, especially given no significant change in translation of the mRNA to produce changes in cellular levels of EGR2 and ARC protein in vivo. Therefore in lines 261-266, the authors need to modify their supposition that exposure may not have been sufficiently long, but rather may already be undergoing recovery from chemical or osmotic insult.
3. That said in comment 2, ARC mRNA (Lines 322-327 and Figure 4) is not significantly different from vehicle. Biological responses are often variable, so no confidence can be placed in statistically nonsignificant results.
4. Lines 311-314: See comments 1 and 2 again with regard to these statements that may well have different explanations.
Author Response
- Abstract lines 17-24, line 358: Certainly astrocytes are not the only type of cell in the brain. However, the finding of downregulation of EGR2 and ARC mRNA in forebrains of whole animals exposed to propylene glycol versus upregulation of EGR2 and ARC mRNA in cultured astrocytes raises yellow flags. This should be better addressed at least in the discussion. I note the differences in concentrations of the different substances to which cultured astrocytes were exposed in vitro in the medium. Dolutegravir exposure is 200 nM, Nicotine and THC 10µM (0.01 mM), but propylene glycol 50 to 150 mM (0.05 to 0.15 M). The propylene glycol concentrations reach the concentration of salt in blood. It is unlikely that propylene glycol would reach these concentrations in the extracellular milieu surrounding brain cells of any kind. Propylene glycol reaches the blood but only after a first pass metabolism in the lungs where an NAD+ dependent alcohol dehydrogenase converts much propylene glycol to lactaldehyde. The authors must comment at least in the discussion that the propylene glycol exposures to cultured astrocytes were very high concentrations and as a consequence, the astrocyte results could be artifactual or a consequence of a cellular chemical or osmotic stress response rather than a specific physiological response to propylene glycol exposure.
- Lines 202-214 and Figure 3C: The initial changes in EGR2 and ARC mRNA may well represent a response such as described in my previous comment from which the cells are undergoing recovery at 18 hours, especially given no significant change in translation of the mRNA to produce changes in cellular levels of EGR2 and ARC protein in vivo. Therefore in lines 261-266, the authors need to modify their supposition that exposure may not have been sufficiently long, but rather may already be undergoing recovery from chemical or osmotic insult.
- Lines 311-314: See comments 1 and 2 again with regard to these statements that may well have differentexplanations.
Response: To comments 1 and 2 and 4: These are important points, which we have now addressed in the Discussion section (page 9, line numbers #325-340; page 8, line numbers #239-247)
“Alternatively, although the concentration of PG was chosen to ensure a robust in-vitro response in order to study the effects of the addition of NIC, THC, and DTG, we concede that a 50-150mM concentration of PG is unlikely comparable to physiologic levels following e-cigarette exposure which are likely not to be more than in the micromolar range 59. Therefore, modulation of EGR2 and ARC in cultured rat astrocytes at 6h and 18h may be the result of a stress response to non-physiologic chemical or osmotic insult rather than a PG-specific effect. Reassuringly, these concentrations of PG did not induce cytotoxicity (data not shown). Both EGR2 and ARC are known to be upregulated under stressful conditions or alterations in metabolic activity 60. Although, both EGR2 and ARC are induced rapidly following stimulus 20,21,58, we suspect that the different pattern seen between EGR2 and ARC at 6h and 18h is likely due to temporal differences in expression induction between EGR2 and ARC, with decreases in ARC at 18h possibly being the result of other homeostatic mechanisms. Future studies are needed to evaluate more physiologically relevant concentrations of PG, and earlier and later time points. However, we find the ability of THC to further enhance this effect and NIC to reduce EGR2 to levels similar to control particularly intriguing.”
“Alterations in gene expression for our validated genes, EGR2 and ARC, did not translate to changes in protein expression (Figure 2c-d) as assessed by Western blot. Although it is possible that the 10-day exposure was not long enough for alternations in mRNA expression to translate to alterations in protein expression, given the previously rapid induction of ARC and EGR2 seen in models of psychoactive substance administration 3230, it is also possible that alterations in post-translational modification are occurring. Post-translational modifications are known to occur and regulate both EGR2 and ARC function 20. Future studies focusing on evaluation of post-translational modifications as well as upstream modulators are needed.”
Further we would like to direct your attention the methods section: “Cells were treated with various combinations of vehicle (DMEM with 5% FBS), propylene glycol (50mM, 100mM, or 150mM, n = 3/group/dose response), Nicotine (10uM), THC (10uM), or Dolutegravir (DTG) (200nM) for 6h or 18h and then processed for RNA” indicating that that these were continuous treatments for 6-18h rather than a 6-18h recovery period following treatment.
- That said in comment 2, ARC mRNA (Lines 322-327 and Figure 4) is not significantly different from vehicle. Biological responses are often variable, so no confidence can be placed instatistically nonsignificant
Response: This is an important point, which we have now addressed in the Discussion section (page 7, line numbers #230-233)
“Typically, it is unwise to draw conclusions from non-significant data. However, because the ARC mRNA levels show a similar overall trend to that of ERG2 we believe that these trends deserve discussion. Therefore, hypothesis and implication regarding both EGR2 and ARC changes are discussed below.”
However, it is unclear to us what reviewer 3 is referring to in regard to “Lines 322-327 and Figure 4” as the paragraphs discussing Figure 4 are limited to EGR2 (please see page 10 and line #345-346) and we discuss the ARC data as follows: “ARC expression changes were subtle and non-significant but followed a similar overall pattern to that of EGR2”.
The conclusion has also been updated to include significant and non-significant wording.
Reviewer 4 Report
Comments and Suggestions for Authors
The present study addresses an important topic in addiction and neuroinflammation by focusing an interplay between
Early growth receptor 2 (EGR2) and Activity Regulated Cytoskeleton Associated 14 Protein (ARC) expression and e-cigarette components
The study focused on two important markers for addiction namely EGR2 and ARC.
The important fact here is the study lacked any behavioral paradigms for addiction, therefore, it is hard to interpret the changes in the markers in the light of the animal behaviors
Another important aspect is the model used, although the model has been validated elsewhere, however I think that adopting a 10-days model for a smoking study is very short, i.e. a lot of markers can be changed, so the question why a longer duration model was not used
Also, the invitro results were challenging for the authors, the acute model perhaps did not reflect the clinical settings for addition, this raises another question : Do the markers studied EGR2 and ARC play a role in early addiction or in later stages?
The Mechanism behind the changes in the makers could be attributed to transcriptional factors? Nrf2 for example?
Also, if the protein levels of EGR2 and ARC was quantified, would the authors think they will have the same findings, i.e. perhaps the proteins were oxidized or underwent post-translational changes?
In conclusion, this work is sound, and it open new horizons for further investigations
Author Response
- The important fact here is the study lacked any behavioral paradigms for addiction, therefore, it is hard to interpret the changes in the markers in the light of the animal behaviors
Response: This is an important point, which we have expanded upon in the Discussion section (page 9, line #292-296):
“However, that conclusion cannot be made based on the current work because behavioral studies were not conducted. However, in support of this hypothesis, a previous study demonstrated that the addition of PG decreased the intracranial self-administration threshold of high-dose nicotine in rats 31.”
- Another important aspect is the model used, although the model has been validated elsewhere, however I think that adopting a 10-days model for a smoking study is very short, i.e. a lot of markers can be changed, so the question why a longer duration model was not used
Response: This is an important point, which we have now addressed in the Methods (page 2, line #82-87) Discussion section (page 10, line #380-385) as follows:
“Two 30-minute sessions were chosen in an attempt to model the multiple per day e-cigarette use seen per day in humans. Nicotine dependence has been shown to develop in as little as 7 to 14-days 11,12.”
“Additionally, although nicotine dependence in rats had been shown to develop in as little as 7-days during nicotine vaporization, and one rat day is equivalent to approximately 30 human days, now that this study has demonstrated acute gene changes fol-lowing 10-days of e-cigarette exposure, more in-depth studies are necessary to deter-mine the effects of acute versus chronic exposure to e-cigarettes”
- Also, the in vitro results were challenging for the authors, the acute model perhaps did not reflect the clinical settings for addition, this raises another question : Do the markers studied EGR2 and ARC play a role in early addiction or in later stages?
Response: This is an important point, which we have now addressed in the Discussion section (page 8, line # 255-260)
“Changes in ARC are induced rapidly, in as little as 30-minutes 31, following psychoactive substance administration, with increases sustained chronically, 24,32 and decreases induced by psychoactive substance withdrawal 24,32. EGR-2 is also induced within hours of psychoactive substance administration, including nicotine administration 29, with increases sustained chronically depending on brain region and treatment paradigm 33,34.”
- The Mechanism behind the changes in the makers could be attributed to transcriptional factors? Nrf2 for example?
Response: This is an important point, which we have now addressed in the Discussion section (page 8, line #270-287)
“It is unclear how inhalation of PG vapor causes decreases in EGR2 and ARC mRNA expression and why the effect is more robust in EGR2. It is possible that the modulation of EGR2 and ARC are not induced directly by PG, NIC, or THC themselves, but that these substances are acting through upstream transcription factors or kinases. EGR2 itself is a transcription factor capable of binding ARC, 37, which may explain the more robust changes seen in EGR2 compared to ARC. ARC is regulated by a variety of upstream factors including cAMP response element-binding protein (CREB), Myocyte Enhancer Factor 2 (MEF2), and Serum response factor (SRF) 38, as well as the kinase Extracellular Signal-regulated Kinases (ERK) 39. Both ARC and EGR2 have also been associated with Nrf2 (Nuclear factor erythroid-2 related factor 2), a transcription factor involved in the regulation of antioxidant and inflammatory responses. Specifically, knockdown of ARC has been demonstrated to inhibit Nrf2 expression. 40 and is increased in parallel to Nrf2 in transgenic animals less prone to oxidative stress 41, whereas parallel increases in EGR2 and Nrf2 are associated with improved retention memory 42. Interestingly, CBD has been shown to modulate Nrf2 and its upstream regulator (Nuclear factor kappa-light-chain-enhancer of activated B cells) NFkB 43, which is also regulates EGR2. Further studies are needed to evaluate to role of upstream transcription factors and other regulatory proteins during the modulation of EGR2 and ARC by PG, NIC, or THC.”
- Also, if the protein levels of EGR2 and ARC was quantified, would the authors think they will have the same findings, i.e. perhaps the proteins were oxidized or underwent post-translational changes?
Response: This is an important point, which we have now addressed in the Discussion section (page 8, line numbers 244-247)
“It is also possible that alterations in post-translational modification are occurring. Post-translational modifications are known to occur and regulate both EGR2 and ARC function 24. Future studies focusing on evaluation of post-translational modifications as well as upstream modulators are needed.”
- In conclusion, this work is sound, and it open new horizons for further investigations
Response: Thank you!